# Effect of TSH stimulation protocols on adequacy of low-iodine diet for radioiodine administration

Hwanhee Lee[1,2], Jin Chul Paeng[1]*, Hongyoon Choi[1], Sun Wook Cho[3], Young Joo Park[3], Do Joon Park[3], Young Ah Lee[4], June-Key Chung[1], Keon Wook Kang[1], Gi Jeong Cheon[1]

1 Department of Nuclear Medicine, Seoul National University College of Medicine, Seoul, Republic of Korea, 2 Department of Nuclear Medicine, Samsung Medical Center, Sungkyunkwan University School of Medicine, Seoul, Republic of Korea, 3 Department of Internal Medicine, Seoul National University College of Medicine, Seoul, Republic of Korea, 4 Department of Pediatrics, Seoul National University Children's Hospital, Seoul, Republic of Korea

* paengjc@snu.ac.kr

**Data Availability Statement:** All relevant data are within the manuscript and its Supporting Information files.

## Abstract

Low-iodine diet (LID) is a crucial preparation for radioactive iodine (RAI) treatment or scan in thyroid cancer. The aim of this study is to analyze the influence of thyroid stimulating hormone (TSH) stimulation protocols and other clinical factors on LID adequacy. Thyroid cancer patients who underwent LID for RAI scan or treatment were retrospectively analyzed. Patients were guided to have LID for 2 weeks before RAI administration and urine iodine/creatinine ratio (UICR, μg/g Cr) was measured. TSH stimulation was conducted using either thyroid hormone withdrawal (THW) or recombinant human TSH (rhTSH) injection. Adequacy of LID was classified by UICR as 'excellent (< 50)', 'adequate (50–100)', 'inadequate (101–250)' and 'poor (> 250)'. A total of 1715 UICR measurements from 1054 patients were analyzed. UICR was significantly higher in case of rhTSH use than THW (72.4 ± 48.1 vs. 29.9 ± 45.8 μg/g Cr, $P < 0.001$). In patients who underwent LID twice using both TSH stimulation protocols alternately, UICR was higher in case of rhTSH than THW regardless of the order of method. Among clinical factors, female, old-age, and the first LID were significant factors to show higher UICR. Although the adequacy of LID was 'adequate' or 'excellent' in most patients, multivariate analysis demonstrated that THW method, male, young age, and prior LID-experience were significant determinants for achieving 'excellent' adequacy of LID. In conclusion, UICR was higher and the proportion of 'excellent' LID adequacy was lower with rhTSH than with THW. UICR was higher also in women, old-age, and LID-naïve patients. Further researches are required to suggest effective methods to reduce body iodine pool in case of rhTSH use and to validate the efficacy of such methods on outcomes of RAI treatment.

**Funding:** This research was supported by grants of the Korea Health Technology R&D Project through the Korea Health Industry Development Institute (KHIDI), funded by the Ministry of Health and Welfare, Republic of Korea (grant number: HI14C1277 and HI18C1916).

**Competing interests:** The authors have declared that no competing interests exist.

## Introduction

In differentiated thyroid cancer, radioactive iodine (RAI) treatment and scan are crucial therapeutic and diagnostic modalities. After total thyroidectomy, remnant thyroid ablation using $^{131}$I is an effective treatment for preventing recurrence in intermediate and high-risk thyroid cancer and high-dose $^{131}$I treatment (usually, $\geq$ 3.7 GBq) is the first treatment of choice in metastatic differentiated thyroid cancer [1,2]. RAI scan using $^{123}$I or $^{131}$I is a sensitive imaging to detect recurrence during follow-up.

RAI scan or treatment requires two kinds of patient preparations; thyroid stimulating hormone (TSH) stimulation and low-iodine diet (LID) [3,4]. TSH stimulation is conducted by thyroid hormone withdrawal (THW) or recombinant human TSH (rhTSH) injection. Recently, rhTSH injection is widely used because it elevates serum TSH level without impairing patients' quality of life [5–7]. LID reduces body iodine pool, which is deemed to enhance RAI uptake by thyroid or cancer cells. Despite some debates on the effect of LID, most guidelines on thyroid cancer recommend LID to restrict iodine intake < 50 μg/day, for 1–2 weeks before RAI administration. However, in case of rhTSH use without THW, metabolized thyroid hormone may be a potential source that increases body iodine pool because iodine is the major component of synthetic thyroid hormones (approximately 64% of levothyroxine and 57% of liothyronine).

Iodine concentration normalized to creatinine level in spot urine (urine iodine/creatinine ratio; UICR) is a reliable index for body iodine pool [8]. In the present study, the adequacy of LID was evaluated using UICR in patients who were prepared for RAI scan or treatment. The aim of the present study is to evaluate the influence of the different TSH stimulation protocols on the adequacy of LID. The influence of various clinical factors on the adequacy of LID were also investigated.

## Materials and methods

### Patients and low-iodine diet

All UICR measurements conducted in Seoul National University Hospital between 2016 and 2020 were retrospectively reviewed. The inclusion criteria were as follows; patients with thyroid cancer who underwent LID for $^{131}$I treatment or $^{123}$I scan between 2016 and 2020, and available UICR measurements conducted for evaluating the adequacy of LID. In patients who underwent more than one LID for repeated RAI scans or treatments, all episodes were included in the analysis. The exclusion criteria were as follows; UICR measurements of children less than 10-year-old at the time of measurement, and outlier results > 1000 μg/g Cr. The design of this study was approved and informed consent from each patient was waived by the Institutional Review Board of Seoul National University Hospital (H2004-181-1119). The electronic medical records of the included patients were accessed from October 2019 to February 2020 and all the data were fully anonymized before analysis.

According to the standard protocol of our institution, all patients were educated on the treatment or scan protocols including diet modification and guided to have LID for 2 weeks before $^{131}$I treatment or $^{123}$I scan. The instruction for LID in our institution is shown in Table 1. Each treatment or scan was scheduled at least 3 months apart from any medical imaging using iodine-containing contrast material. TSH stimulation was conducted by either THW or rhTSH. In the THW protocol, thyroid hormone replacement was stopped for 4 weeks, and was substituted by liothyronine during the first 2 weeks. In the rhTSH protocol, thyroid hormone replacement was continued except only the day of RAI administration and rhTSH (0.9 mg) was injected intramuscularly twice, one and two days before RAI administration.

**Table 1. Instructions of LID for RAI treatment or scan.**

| Food | Instructions | |
|---|---|---|
| | Allowed | Restricted |
| Grains | Rice, flour, potato and sweet potato without peel | Noodles with sock including kelp, anchovy, etc. Bread including milk or egg yolks |
| Meat, fish, dairy products | Beef, pork, chicken (Restricted less than 120 g/day) | All seafood (fish, shrimp, crab, oysters, etc.) Processed meat Egg yolk, milk and dairy products |
| Vegetables, fruits, legumes | Most vegetables, fruits, nuts | All seaweed (laver, sea mustard, green algae, kelp, etc.) Salted vegetables Canned/bottled/concentrated processed fruits/vegetables Tofu with natural coagulant |
| Seasonings | Refined salt, sugar, red pepper, catsup, vinegar | See salt, imported ionized salt Soybean/red pepper paste containing sea salt Mayonnaise |
| Others | Coffee, tea | Iodine-containing vitamins or food supplements Foods and drugs containing red food dyes Chocolate |

## Urine iodine measurement and clinical data acquisition

Spot urine and blood samples were obtained in the morning of the day of RAI administration. Urine iodine concentration was measured by inductively coupled plasma mass spectrometry, and urine creatinine was measured by colorimetric assay. Urine iodine level was normalized to the creatinine concentration to yield UICR that is expressed as µg/g Cr. UICR higher than 1000 µg/g Cr was deemed to be a measuring error and excluded from the analysis. Adequacy of LID was classified as excellent, adequate, inadequate, and poor, when UICR was < 50 µg/g Cr, 50–100 µg/g Cr, 101–250 µg/g Cr, and > 250 µg/g Cr, respectively [9]. Data on demographic and clinical factors were obtained from review of electronic medical records.

## Statistical analysis

Results were shown as mean ± standard deviation, and interquartile range (IQR) was also shown due to some outlier data. In comparison of UICR values, $t$-test was used for two-group comparisons (THW vs. rhTSH; male vs. female; ILD-experienced vs. ILD-naïve) and ANOVA was used for multiple-group comparison (in terms of clinical factors). In comparison of ordinal variables like UICR adequacy, Mann-Whitney $U$ test and Kruskal-Wallis H test were used for two-group and multiple-group comparisons, respectively. Wilcoxon signed-rank test was used to compare repeated UICR measurements of a same patient. Multivariate analysis using stepwise regression was performed to determine independent and significant factors for achieving 'excellent' adequacy of LID, with including variables that were significant in univariate analyses. Risk was calculated as odds ratio (OR) and 95% confidence interval (CI) was obtained. Data were analyzed by a commercial statistics software package (MedCalc, ver.15.8, MedCalc Software bvba., Ostend, Belgium) and $P < 0.001$ was regarded statistically significant.

## Results

### Patients

A total of 1054 patients (M:F = 318:736, age 46.7±14.6 y, range 11–89 y) were included in the analysis; 1045 (99.1%) were differentiated thyroid cancer, and 9 (0.9%) were poorly

**Table 2. Patient characteristics.**

| Characteristics | N (%) |
|---|---|
| Sex | |
| Male | 318 (30.2%) |
| Female | 736 (69.8%) |
| Age | |
| < 30 | 133 (12.6%) |
| 30 ~ 50 | 481 (45.6%) |
| 50 ~ 70 | 367 (34.8%) |
| ≥ 70 | 73 (6.9%) |
| Pathology | |
| Papillary | 985 (93.5%) |
| Follicular | 60 (5.7%) |
| Poorly differentiated | 9 (0.9%) |
| Number of LID experience* | |
| 1 | 267 (25.3%) |
| 2 | 581 (55.1%) |
| ≥ 3 | 206 (19.5%) |

*Including LID experience before study period.

differentiated thyroid cancer. During the study period, 502 (47.6%) patients underwent LID once and 552 (52.4%) patients underwent LID more than once. Patients' demographic and clinical data are summarized in Table 2. A total of 1718 LID and urine iodine measurement were performed during the study period, and 3 UICR measurements were > 1000 μg/g Cr and excluded from the analysis. Finally, 1715 measurements were analyzed; of which 392 (22.9%) were performed for $^{123}$I scan, 401 (23.4%) were for low-dose (< 1.2 GBq) $^{131}$I treatment, and 922 (53.8%) were for high-dose (≥ 1.2 GBq) $^{131}$I treatment.

## Urine iodine level according to TSH stimulation protocol

Mean UICR of 1715 measurements was 50.6 ± 51.5 μg/g Cr (median 42.1, IQR 18.6–65.0, range 1.6–643.4 μg/g Cr). Adequacy of LID was excellent in 990 (57.7%), adequate in 599 (34.9%), inadequate in 110 (6.4%), and poor in 16 (0.9%).

For TSH stimulation, THW was used in 880 (51.3%), and rhTSH was used in 835 (48.7%) cases. In overall cases, UICR was significantly higher in rhTSH group than in THW group (72.4 ± 48.1 vs. 29.9 ± 45.8 μg/g Cr, $P < 0.001$). In adequacy assessment, 779 (88.5%) were 'excellent' in THW group, whereas the majority cases (n = 524, 62.8%) were 'adequate' in rhTSH group ($P < 0.001$, Fig 1A).

Of the patients who underwent LID twice or more, 514 patients underwent his/her first-time and second-time LID during the study period (Table 3). Among them, those who used the same protocol showed only a minor difference in the LID adequacies of the first and second LID (Fig 1B). However, in 243 patients who experienced both THW and rhTSH by crossed protocols, the proportion of 'excellent' adequacy was higher in THW cases, regardless of the order of protocols ($P < 0.001$). In these patients, UICR was significantly higher in case of rhTSH than THW, in both rhTSH-first group (72.8 ± 39.6 vs. 29.9 ± 34.5, $P < 0.001$) and THW-first group (70.0 ± 62.1 vs. 26.8 ± 18.9, $P < 0.001$).

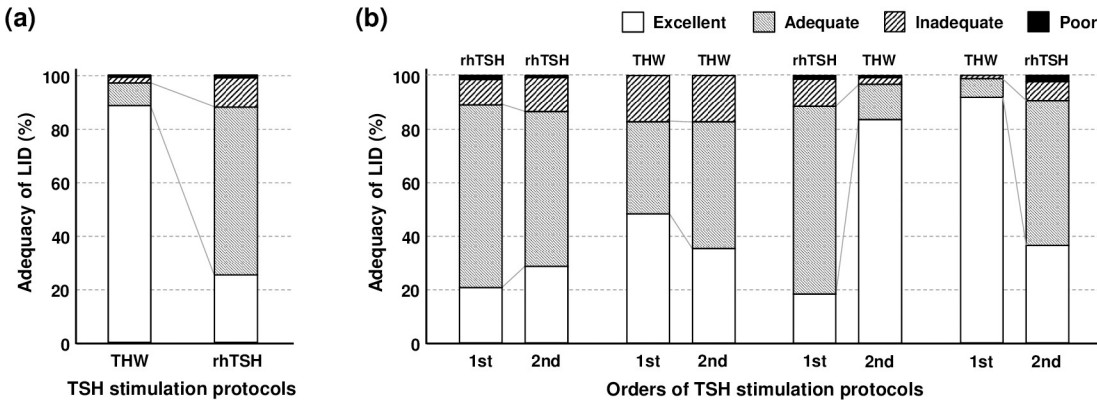

**Fig 1. Adequacy of LID according to TSH stimulation protocols.** In overall LID, THW group showed a higher proportion of 'excellent' adequacy than rhTSH group (a). In patients who underwent the first and second LID during the study period, the proportion of 'excellent' adequacy was higher in THW cases regardless of the order of protocols (b).

### Urine iodine level according to clinical factors

UICR was significantly higher in women than in men ($55.0 \pm 52.8$ vs. $40.4 \pm 47.1$, $P < 0.001$, Table 4). Although 95.1% of men and 91.6% of women were in 'excellent' or 'adequate' range of LID, the proportion of 'excellent' LID was higher in men (74.0% vs. 50.5%, $P < 0.001$, Fig 2A). UICR was increased with age and there were significant differences between the age groups ($35.0 \pm 31.1$, $45.9 \pm 47.2$, $56.1 \pm 50.3$, and $83.3 \pm 86.6$ for each age group, respectively, $P < 0.001$, Table 4). In patients younger than 30 years, only 3.4% were 'inadequate', whereas 20.0% were 'inadequate' or 'poor' in patients of age $\geq$ 70 years (Fig 2B). UICR was significantly higher in the first LID than that in second or more LID ($55.3 \pm 44.2$ vs. $46.6 \pm 56.6$, $P < 0.001$). In terms of procedures, UICR in low-dose $^{131}$I treatment were lower than those of $^{123}$I scan or high-dose $^{131}$I treatment, although the difference was not significant between $^{123}$I scan or high-dose $^{131}$I treatment (Table 4).

### Multivariate analysis

Multivariate analysis was performed to determine significant determinants for achieving 'excellent' adequacy class of LID, including variables of sex (male vs. female), age ($< 50$ vs. $\geq$

**Table 3. The number of patients according to TSH stimulation protocols and LID procedures in those who underwent the first and second LID during the study period.**

| Protocols | $^{123}$I scan | $^{131}$I ($< 1.2$ GBq) | $^{131}$I ($\geq 1.2$ GBq) | Sum |
|---|---|---|---|---|
| rhTSH–rhTSH | | | | |
| 1st LID | 2 | 22 | 102 | 126 |
| 2nd LID | 64 | 43 | 19 | 126 |
| THW–THW | | | | |
| 1st LID | 3 | 12 | 130 | 145 |
| 2nd LID | 58 | 37 | 50 | 145 |
| rhTSH–THW | | | | |
| 1st LID | 0 | 38 | 120 | 158 |
| 2nd LID | 18 | 93 | 47 | 158 |
| THW–rhTSH | | | | |
| 1st LID | 1 | 1 | 83 | 85 |
| 2nd LID | 73 | 2 | 10 | 85 |

**Table 4. UICR according to demographic and clinical factors.**

| Factors | | UICR (µg/g Cr) | | | *P* |
|---|---|---|---|---|---|
| | | **Overall** | **rhTSH** | **THW** | |
| Sex | | | | | < 0.001 |
| Men | n | 526 | 287 | 239 | |
| | value | 40.4 ± 47.1 | 26.4 ± 49.1 | 57.4 ± 38.1 | |
| Women | n | 1189 | 593 | 596 | |
| | value | 55.0 ± 52.8 | 31.6 ± 44.1 | 78.3 ± 50.3 | |
| Age (y) | | | | | < 0.001 |
| < 30 | n | 232 | 144 | 88 | |
| | value | 35.0 ± 31.1 | 18.9 ± 15.2 | 61.3 ±32.5 | |
| 30~50 | n | 768 | 398 | 370 | |
| | value | 45.9 ± 47.2 | 26.0 ± 41.2 | 67.2 ± 43.8 | |
| 51~70 | n | 593 | 295 | 300 | |
| | value | 56.1 ± 50.3 | 36.1 ± 49.3 | 75.8 ± 43.0 | |
| > 70 | n | 120 | 43 | 77 | |
| | value | 83.3 ± 86.6 | 60.3 ± 91.2 | 96.0 ± 81.7 | |
| LID experience | | | | | < 0.001 |
| No (first) | n | 784 | 316 | 468 | |
| | value | 55.3 ± 44.2 | 28.3 ± 29.1 | 73.5 ± 43.3 | |
| Yes (≥ second) | n | 931 | 567 | 367 | |
| | value | 46.6 ± 56.6 | 30.8 ± 53.0 | 70.8 ± 53.5 | |
| Procedures | | | | | 0.007 |
| $^{123}$I scan | n | 392 | 145 | 247 | |
| | value | 55.6 ± 60.3 | 29.9 ± 57.3 | 70.7 ± 56.9 | |
| $^{131}$I (< 1.2 GBq) | n | 401 | 231 | 170 | |
| | value | 45.1 ± 44.1 | 27.5 ± 41.1 | 68.9 ± 36.2 | |
| $^{131}$I (≥ 1.2 GBq) | n | 922 | 504 | 418 | |
| | value | 50.8 ± 50.2 | 31.0 ± 44.2 | 74.7 ± 46.6 | |

50 years), prior LID experience (yes vs. no), procedure ($^{123}$I scan vs. $^{131}$I treatment) and TSH stimulation protocol (THW vs. rhTSH). In univariate analyses, female, old age, no prior LID experience and rhTSH use were strongly associated with higher UICR ($P = 0.013$ for

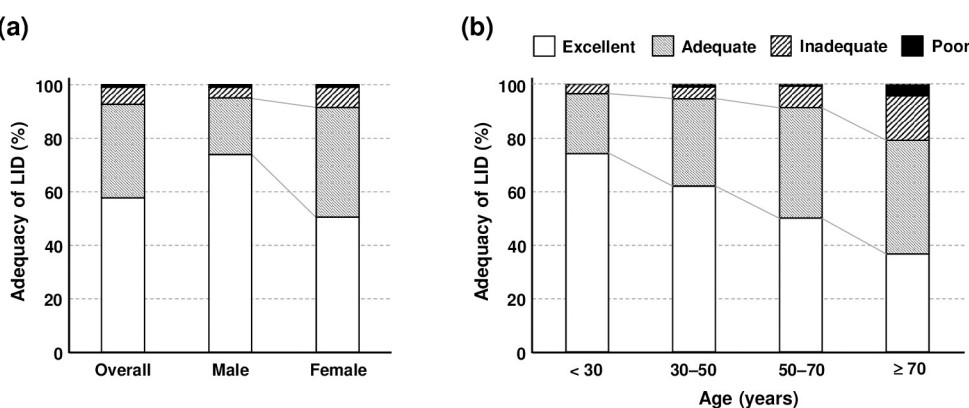

**Fig 2. Adequacy of LID according to sex (a) and age (b).** Male and young patients showed higher proportion of 'excellent' adequacy.

**Table 5. Multivariate analysis for determinants to achieve 'excellent' LID adequacy.**

| Parameters | Classification | Univariate | | Multivariate | |
|---|---|---|---|---|---|
| | | OR (95% CI) | P | OR (95% CI) | P |
| Sex | Male vs. female | 2.78 (2.22–3.48) | < 0.001 | 3.63 (2.70–4.89) | < 0.001 |
| Age | < 50 vs. ≥ 50 years | 2.01 (1.65–2.44) | < 0.001 | 2.51 (1.91–3.28) | < 0.001 |
| LID experience | Yes vs. no | 2.06 (1.70–2.50) | < 0.001 | 1.41 (1.08–1.83) | 0.011 |
| Procedure | Treatment vs. scan | 1.33 (1.06–1.67) | 0.013 | - | n.s.* |
| TSH stimulation protocol | THW vs. rhTSH | 22.81 (17.60–29.57) | < 0.001 | 27.25 (20.44–36.34) | < 0.001 |

* Not selected as significant.

procedures, and $P < 0.001$ for others; Table 5). In the multivariate analysis, procedure was not selected as a significant factor, and all the other factors were significant. In particular, OR of TSH stimulation protocol was 27.25 (95% CI, 20.44–36.34), which was the highest among OR values of all other factors (Table 5).

## Discussion

In the present study, UICR was analyzed in a large cohort including 1715 measurements of 1054 patients, and it was demonstrated that UICR is significantly associated with TSH stimulation protocol. In most cases (92.7%), 2-week LID resulted in 'excellent' or 'adequate' range of UICR. However, UICR was higher and proportion of 'excellent' adequacy was lower in patients who used rhTSH without THW compared to those used THW. In patients who underwent repeated LID using both rhTSH and THW protocols, UICR was higher in case of rhTSH, regardless of the order of used protocols. Furthermore, TSH stimulation protocol was a significant and independent determinant for achieving 'excellent' adequacy of LID.

Reduction of body iodine pool is expected to enhance the effect of RAI treatment, by reducing competition between RAI and body iodine for target tissue uptake. However, there have been some debates on the effect of lowering body iodine pool on RAI treatment. In an early study, no significant difference was observed in ablation success rates between patients on 2-week LID and those on regular diet [10]. In another study, urine iodine excretion was also not related to ablation success rate [11]. In contrast, several other studies suggested potential effect of LID, such as low urine iodine level and high 24-hour neck RAI uptake in LID [12]. In a relatively recent study, ablation success rate was 50% in patients with UICR ≥ 250 μg/g Cr whereas it was 76–82% in patients with UICR < 50 μg/g Cr, although the ablation success rates of those with UICR < 250 μg/g Cr and those with < 50 μg/g Cr were similar [13]. In a systematic review, LID was recommended based on its effect to reduce urine iodine level [14], despite lack of evident data on long-term outcome or mortality.

Currently, major guidelines on thyroid cancer recommend LID before remnant ablation or RAI scan, although there is no consensus on the optimal duration of LID. In a comparison study between 2-week and 3-week LID protocols, both resulted in significant decrease in urine iodine level and no difference was observed between them [15]. In other studies, 4-day, 1-week and even 4-week LID protocols have been used to attain satisfactory urine iodine level [9,12,16]. While the American Thyroid Association and the European Association of Nuclear Medicine recommend 1 to 2-week LID prior to RAI administration, the European Thyroid Cancer Taskforce recommends 3-week LID [1,3,17,18].

For TSH stimulation in RAI scan or treatment, rhTSH has been increasingly used to avoid hypothyroidism symptoms without inferiority in ablation efficacy [5–7]. Use of rhTSH can affect iodine metabolism in two ways; preserved renal iodine clearance and increase in iodine

supply. When rhTSH is used without THW, euthyroid state is maintained and renal function of iodine clearance is preserved [19,20]. Although high renal clearance of RAI is helpful to reduce radiation exposure of normal tissues, it may attenuate efficacy of RAI treatment. Furthermore, continued thyroid hormone replacement can be a potential source of body iodine. Approximately 100 μg/day of levothyroxine would be equivalent to 64 μg/day of additional dietary iodine. In an early study on patients who used rhTSH without THW, adequate urine iodine level was attained in 71% of patients with 2-week LID, and the rate was only 41% with 1-week LID [9]. In another study including 201 patients, urine iodine level was higher in those who used rhTSH without THW than those who used rhTSH with 4-day THW. Performance of ablation was also higher in those who used rhTSH with 4-day THW [21].

The main finding of the present study is in line with the previous results that continued thyroid hormone replacement with rhTSH use can affect body iodine. Although most of patients were within 'excellent' or 'adequate' range of UICR with rhTSH, there was a significant difference in UICR values between THW and rhTSH groups. This difference may affect efficacy of RAI treatment particularly in those who need maximal efficacy of RAI treatment (such as high-risk patients) or those who are prone to poor LID adequacy (such as old patients). Thus, further researches are warranted to find out effective methods to reduce body iodine pool in case of rhTSH use. Additionally, the efficacy of such methods on outcomes of RAI treatment needs to be evaluated.

In the present study, sex, age, and LID experience were also significant determinants for attaining 'excellent' adequacy of LID. The influence of sex and age may have been partly caused by the characteristics of UICR measurement. High muscle mass and urinary excretion of creatinine in male or young patients would cause low UICR ratio. In a study on normal population, UICR was reported to be generally lower in men than in women [22], and it was suggested that age- and sex-adjusted UICR may be a better index than simple UICR [23]. The influence of age and LID experience may be caused by the differences in level of understanding and compliance of patients. Younger or experienced patients would understand guidance for LID and comply with it, better than older or LID-naïve patients. The effect of procedures on LID can also be caused by compliance. Low-dose [131]I treatment and [123]I scan were often performed at second or more times of LID. Additionally, patients' compliance with LID may have been better in case of [131]I treatment than [123]I scan.

There are a few limitations in the present study. First, this study is retrospective and it was unavailable to assess patients' compliance with LID although standard education program on LID and RAI treatment was provided to every patient. Second, rhTSH use was not systematically determined. In our medical insurance system, rhTSH is covered by the National Health Insurance Services only once in lifetime for ablation and follow-up examination, respectively. Thus, rhTSH was used by physicians' decision and/or patients' preference. Third, LID for RAI scan, ablation, and treatment were analyzed together. Although preparation protocols are same for all these procedures in our institution, there may have been some unexpected bias such as patients' compliance with LID. However, there was not a significant interference of procedures in the multivariate analysis. Another minor limitation of the study is the use of UICR as a marker for body iodine pool, because urine iodine measured from 24-hour urine is an effective marker for body iodine pool [24]. However, UICR from spot urine has been widely used due to difficulty and inconvenience of 24-hour urine collection. UICR has a close correlation with total iodine contents measured from 24-hour urine [8]. Finally, differences in outcomes of ablation or treatment were not analyzed in this study. Because the main purpose of this study was to compare LID efficacy between two different TSH stimulation protocols, all repeated scans and RAI treatments were included in this study.

## Conclusions

After 2-week LID, UICR is reduced to 'excellent' or 'adequate' level in majority of patients. However, UICR is higher and the proportion of 'excellent' LID adequacy is lower with rhTSH than with THW. UICR is higher also in women, old-age, and LID-naïve patients. Because high body iodine pool may affect the efficacy of RAI treatment, further researches are required for effective methods to reduce body iodine pool in case of rhTSH use, and the efficacy of such methods on outcomes of RAI treatment.

## Supporting information

**S1 Dataset.**
(XLSX)

## Author Contributions

**Conceptualization:** Hwanhee Lee, Jin Chul Paeng, Young Joo Park.

**Data curation:** Hwanhee Lee, Jin Chul Paeng.

**Formal analysis:** Hwanhee Lee, Jin Chul Paeng, Young Ah Lee.

**Funding acquisition:** Jin Chul Paeng.

**Investigation:** Hwanhee Lee, Jin Chul Paeng.

**Methodology:** Hwanhee Lee, Jin Chul Paeng, Young Ah Lee.

**Project administration:** Jin Chul Paeng.

**Resources:** Hwanhee Lee, Jin Chul Paeng, Sun Wook Cho, Young Joo Park, Do Joon Park, June-Key Chung, Keon Wook Kang, Gi Jeong Cheon.

**Software:** Jin Chul Paeng.

**Supervision:** Jin Chul Paeng, June-Key Chung, Keon Wook Kang.

**Validation:** Hwanhee Lee, Jin Chul Paeng.

**Visualization:** Hwanhee Lee, Jin Chul Paeng.

**Writing – original draft:** Hwanhee Lee.

**Writing – review & editing:** Hwanhee Lee, Jin Chul Paeng, Hongyoon Choi, Sun Wook Cho, Young Ah Lee.

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
