## [Decision Letter · Decision Letter 0]

25 May 2021

PONE-D-21-02040

Effect of TSH Stimulation Protocols on Adequacy of Low-Iodine Diet for Radioiodine Administration

PLOS ONE

Dear Dr. Paeng,

Thank you for submitting your manuscript to PLOS ONE. After careful consideration, we feel that it has merit but does not fully meet PLOS ONE’s publication criteria as it currently stands. Therefore, we invite you to submit a revised version of the manuscript that addresses the points raised during the review process.

We look forward to receiving your revised manuscript.

Kind regards,

Ali S. Alzahrani

Academic Editor

PLOS ONE

Journal Requirements:

2. In the ethics statement in the manuscript and in the online submission form, please provide additional information about the patient records/samples used in your retrospective study, including: a) whether all data were fully anonymized before you accessed them; b) the date range (month and year) during which patients' medical records/samples were accessed.

Reviewers' comments:

Reviewer's Responses to Questions

**Comments to the Author**

1. Is the manuscript technically sound, and do the data support the conclusions?

Reviewer #1: Yes

Reviewer #2: Partly

2. Has the statistical analysis been performed appropriately and rigorously? 

Reviewer #1: Yes

Reviewer #2: No

3. Have the authors made all data underlying the findings in their manuscript fully available?

Reviewer #1: No

Reviewer #2: No

4. Is the manuscript presented in an intelligible fashion and written in standard English?

Reviewer #1: Yes

Reviewer #2: Yes

5. Review Comments to the Author

Reviewer #1: This is a well-deigned and well-written retrospective analysis in a large cohort to analyze effect of different clinical factors including the two TSH stimulation protocol on the LID utilizing the spot UICR.

Purpose of the study is important and impactful taking into consideration the increasing use of rhTSH and more practical spot UICR. Methodology is systematic, well-designed and clear; moreover, it can be replicated in future analysis to find out the impact on recurrence and survival that were not addressed in this study. Discussion is comprehensive and it covers the relevant literature, strength and weakness of current manuscript and the potential future directions and impact of current findings. Conclusion is well drawn based on the presented interesting data. However, it will be highly impactful if data on survival presented or to consider this by authors in future follow up work.

Just minor correction of thyroid globulin in the introduction (to replace it by thyroglobulin TG)

Reviewer #2: Overall

Thank you for this clearly written manuscript. It describes a retrospective review of routinely collected clinical data in patients with differentiated thyroid cancer scheduled for radioiodine (RAI) scan or treatment in South Korea. Patients were prepared for RAI with either recombinant human thyroid stimulating hormone injection (rhTSH) or thyroid hormone withdrawal (THW) to raise TSH and were advised to follow a 2-week low iodine diet. Urine iodine creatinine ratio was measured on the day of RAI. The authors report that iodine levels are higher in patients prepared with rhTSH and recommend that, for patients undergoing RAI for treatment purposes, short term THW should be considered alongside rhTSH. Analyses of other clinical factors were undertaken.

The study is probably the largest study reporting on urine iodine status prior to RAI to date and the authors should be commended for that. Findings that patients prepared with rhTSH have a higher iodine status after 2-week dietary restriction than those prepared with THW are in line with other studies in this area. However, the study did not analyse the outcome of ablation/treatment, and this limits the recommendations that can be made. There are very few studies that have found any association between urinary iodine levels and treatment success and those that exist are subject to bias due to lack of control of covariates. There is a real need for better evidence for using LIDs which are an additional treatment burden for patients, but this study barely addresses this issue.

I recommend publication but there are improvements that should be made.

Major

Statistical analysis

If the data are normally distributed as implied by the use of means and standard deviations, non-parametric tests should not be used and a re-analysis using t-tests and ANOVAs is necessary. (If the data are not normally distributed, please report medians instead of the means).

There needs to be more specific details given about the comparisons being made (eg line 100 – what two groups; line 103 what repeated measurements?)

I am finding it difficult to understand the explanation of the multivariate analysis. Exactly which variables were entered and why?

Results

A comparison of urinary iodine in patients prepared with rhTSH vs THW was, I think, the primary research question (and seems to be the most important factor in whether or not someone achieved excellent depletion). Therefore, it seems odd to present the urine iodine levels according to demographic and clinical factors first and completely independently of preparation. I suggest “urine iodine levels according to TSH stimulation protocol” be presented immediately after the demographics. The comparison of demographic and clinical factors can then be presented. Information should be given on the number of patients in each clinical group who were prepared with rhTSH vs THW and the corresponding mean/median urinary iodine measures (add this to table 2).

Discussion and conclusion

The authors should be much more cautious in recommending that short-term THW be considered alongside rhTSH because this study has not demonstrated that the proposed protocol would effectively lower urinary iodine more than rhTSH alone, or that there would be any impact on success rates. It could be a recommendation for future research.

Some evidence is presented from Barbero et al, a study, that did analyse treatment outcomes between various preparatory protocols (although the study used furosemide, which is not usually recommended) to support the recommendation. However, the authors should also consider and discuss evidence from Tala Jury et al (citation 12) that reported “…no significant difference in UIE between ablated or nonablated patients both in the whole group and the rhTSH or THW groups” and from the HiLo and ESTIMABLE1 trials, large RCTs that found no evidence for a difference in outcomes at 6-8 months and 5-year follow-up, between patients prepared with rhTSH vs THW.

Line 238. Rather than recommending an intervention that they haven’t tested which is more disruptive for patients, could the authors comment on whether improved dietary advice for those receiving rhTSH would achieve the desired results?

Given that evidence that very low iodine status enhances treatment success is very weak, I would like to see an explanation of why no analysis was made of the outcome of ablation or treatment during this study. As this is a retrospective study the data should be available.

The abstract would need re-writing to reflect these comments.

Minor

Introduction

Define intermediate and high-risk thyroid cancer and high-dose 131I.

Line 47-49. Remove, this observation is not relevant to this paper.

It would be helpful to set out the research question(s) much more clearly and to provide some explanation as to why the analyses of other clinical factors were undertaken.

Methods.

Please provide a full description of the inclusion and exclusion criteria.

There are substantial variations in the LIDs advised between different countries and institutions. It would be useful to include a table showing the advised LID used in this study.

Statistical analysis.

Statistical significance is a function of sample size – it is better to give the precise p-values (which has been done here), to talk about levels of evidence instead of referring to a result as ‘significant’ and to consider the effect size (see https://www.ncbi.nlm.nih.gov/pmc/articles/PMC1119478/).

Results (although may not all be relevant after addressing the major recommendations)

Lines 113 – 116: Cut text from 985 to 2 times. All of this is presented in table 1.

Table 1 – Please give information on staging.

Line 129, 132, 133, 136. Quote the effect not the p-values in the text. For example, line 129 “There was strong evidence that UICR was higher in women than in men (55.0 (52.8) vs 40.4 (47.1) ug/gCr)” is much more meaningful than just quoting the p-value.

Retitle table 2 and the corresponding heading ‘according to demographic and clinical factors’.

Include number of patients for each factor.

Consider removing the figures and putting the information into table 2.

Line 148: Report numbers achieving ‘excellent’ depletion for both groups.

The title of table 3 doesn’t match what the table is showing. Also, it needs to be clearer that the numbers in each cell in table 3 refers to numbers of patients.

Discussion

Please start the discussion with the main findings, not information about whether urine iodine creatinine is a good measure (that can go under limitations).

Line 200. “….body retention of RAI increases with use of LID”. This effect has been observed elsewhere (Maruca et al., 1984) and is not desirable (although Plujimen et al do not consider it to be highly problematic). Is this what the authors wanted to communicate from this paper?

Line 202. Please amend to show the study findings more completely ie that success rate was similar (about 80%) for all patients with UICR <250mg/gCr, not just those <50mcg/gCr. Sohn et al cannot be used to suggest that a UICR <50mcg/gCr is superior to <100mcg/gCr.

Line 211 The European Thyroid Cancer Taskforce made that recommendation in 2006 and it does not reflect guidelines from other European bodies which mainly advise 1-2 weeks (eg European Association of Nuclear Medicine in 2008 recommended 1-2 weeks (Luster et al) – please add this reference. (And Pluijmen et al, cited as a study that demonstrated LIDs improve ablation only used 4 days – perhaps comment on that).

Line 241 – 252. There is some discussion of the demographic and clinical differences, can the authors comment on whether changes to how dietary advice is delivered locally could help with these differences.

I have not extensively proof-read the article or corrected minor errors in English but there are some and I suggest a proof reader be employed.

6. PLOS authors have the option to publish the peer review history of their article (what does this mean?). If published, this will include your full peer review and any attached files.

Reviewer #1: **Yes: **Akram Al-Ibraheem

Reviewer #2: **Yes: **Clare England

---

## [Author Response · Author response to Decision Letter 0]

18 Jun 2021

Dear Dr. Alzahrani,

Thank you for giving us an opportunity to revise our paper on ‘Effect of TSH Stimulation Protocols on Adequacy of Low-Iodine Diet for Radioiodine Administration’ to PLOS ONE. We appreciate you and the reviewers taking time and effort to provide valuable feedback on the manuscript. We are grateful to the reviewers for their insightful comments on the manuscript and the suggestions have been immensely helpful. We have highlighted in red the changes within the marked-up copy of our manuscript as you requested. 

Regarding the journal requirements, we have added information about patient records used in out retrospective study and uploaded the study’s minimal underlying data set file as Supporting information files. We apologize for neglecting that requirement in PLOS data policy when we originally submitted the manuscript.

We hope the revised manuscript will better suit the PLOS ONE but are happy to consider further revisions, and we thank you for your continued interest in our study.

Response to Reviewers:

First of all, we appreciate all the reviewers’ helpful comments. We have made changes to the manuscript according to the reviewers’ comments. Below, please find our point-by-point responses to the comments. The reviewers’ comments are in blue italics, and our responses are in regular text, with changes to the manuscript in red. We hope that our revised manuscript is suitable for publication in the PLoS One.

(We changed the order of Figs. 1 and 2 according to the changes in main text.)

Reviewer #1

1. Just minor correction of thyroid globulin in the introduction (to replace it by thyroglobulin TG)

We appreciate the reviewer’s comment and corrected the error.

Reviewer #2

Major

[Statistical Analysis]

1. If the data are normally distributed as implied by the use of means and standard deviations, non-parametric tests should not be used and a re-analysis using t-tests and ANOVAs is necessary.

As the reviewer commented, ‘UICR level’ was a continuous variable and showed normal distributions. In our previous manuscript, statistical methods used for comparing UICR levels were missing, and we added them. Non-parametric tests were used for ordinal variables.

(Page 5, line 121-126) In comparison of UICR values, t-test was used for two-group comparisons (THW vs. rhTSH; male vs. female; ILD-experienced vs. ILD-naïve) and ANOVA was used for multiple-group comparison (in terms of clinical factors). In comparison of ordinal variables like UICR adequacy, Mann-Whitney U test and Kruskal-Wallis H test were used for two-group and multiple-group comparisons, respectively.

2. There needs to be more specific details given about the comparisons being made (eg. line 100 – what two groups; line 103 what repeated measurements?)

We revised the manuscript to describe more specifically the comparisons that we performed. Please refer to our response to the above comment 1.

3. I am finding it difficult to understand the explanation of the multivariate analysis. Exactly which variables were entered and why?

We attempted to determine which factor is independently significant to achieve ‘excellent’ adequacy class of LID. We included such variables that are significant for determining ‘excellent’ adequacy of LID in univariate analyses. Finally, sex (male vs. female), age (< 50 vs. � 50 years), prior LID experience (yes vs. no), procedure (123I scan vs. 131I treatment) and TSH stimulation protocol (THW vs. rhTSH) were included in the multivariate analysis. We clarified the method and included variables in the manuscript.

(Page 5, line 127-129) Multivariate analysis using stepwise regression was performed to determine independent and significant factors for achieving ‘excellent’ adequacy of LID, with including variables that were significant in univariate analyses.

(Page 10, line 200-203) Multivariate analysis was performed to determine significant determinants for achieving ‘excellent’ adequacy class of LID, including variables of sex (male vs. female), age (< 50 vs. � 50 years), prior LID experience (yes vs. no), procedure (123I scan vs. 131I treatment) and TSH stimulation protocol (THW vs. rhTSH).

[Results]

4. A comparison of urinary iodine in patients prepared with rhTSH vs THW was, I think, the primary research question (and seems to be the most important factor in whether or not someone achieved excellent depletion). Therefore, it seems odd to present the urine iodine levels according to demographic and clinical factors first and completely independently of preparation. I suggest “urine iodine levels according to TSH stimulation protocol” be presented immediately after the demographics. The comparison of demographic and clinical factors can then be presented.

We changed the order of paragraphs according to the reviewer’s recommendation.

(Page 7–10) 

5. Information should be given on the number of patients in each clinical group who were prepared with rhTSH vs THW and the corresponding mean/median urinary iodine measures (add this to table 2).

We added the data, according to the reviewer’s comment.

(Table 4) UICR according to demographic and clinical factors

[Discussion and conclusion]

6. The authors should be much more cautious in recommending that short-term THW be considered alongside rhTSH because this study has not demonstrated that the proposed protocol would effectively lower urinary iodine more than rhTSH alone, or that there would be any impact on success rates. It could be a recommendation for future research. Some evidence is presented from Barbero et al, a study, that did analyse treatment outcomes between various preparatory protocols (although the study used furosemide, which is not usually recommended) to support the recommendation. However, the authors should also consider and discuss evidence from Tala Jury et al (citation 12) that reported “…no significant difference in UIE between ablated or nonablated patients both in the whole group and the rhTSH or THW groups” and from the HiLo and ESTIMABLE1 trials, large RCTs that found no evidence for a difference in outcomes at 6-8 months and 5-year follow-up, between patients prepared with rhTSH vs THW. Line 238. Rather than recommending an intervention that they haven’t tested which is more disruptive for patients, could the authors comment on whether improved dietary advice for those receiving rhTSH would achieve the desired results?

We agree with the reviewer. Although the UICR level and achieved LID adequacy were better with THW than with rhTSH, additional evidence is required regarding the effect of short-term THW on UICR level or RAI treatment outcome. According to the reviewer’s comment, we revised the manuscript as shown below. However, additional diet modification is difficult in patients with rhTSH use, because instruction for iodine restriction diet is already considerably strict in our institution and many hospitals.

(Page 13, line 260-268) The main finding of the present study is in line with the previous results that continued thyroid hormone replacement with rhTSH use can affect body iodine. Although most of patients were within ‘excellent’ or ‘adequate’ range of UICR with rhTSH, there was a significant difference in UICR values between THW and rhTSH groups. This difference may affect efficacy of RAI treatment particularly in those who need maximal efficacy of RAI treatment (such as high-risk patients) or those who are prone to poor LID adequacy (such as old patients). Thus, further researches are warranted to find out effective methods to reduce body iodine pool in case of rhTSH use. Additionally, the efficacy of such methods on outcomes of RAI treatment needs to be evaluated.

7. Given that evidence that very low iodine status enhances treatment success is very weak, I would like to see an explanation of why no analysis was made of the outcome of ablation or treatment during this study. As this is a retrospective study the data should be available.

We agree with the reviewer that the treatment efficacy should be the final endpoint. However, our study was initially focused on the effect of TSH stimulation protocols on UICR, and designed to include somewhat heterogeneous situations including I-123 scan, RAI ablation and treatment of various doses. Additionally, repeated scans or RAI treatments were all included. Thus, relevant analyses on the ablation success rate or RAI treatment efficacy were not evaluated in our study. As the reviewer commented, we mentioned that the treatment efficacy should be investigated in future researches, and we also mentioned it as a limitation of our study.

(Page 13, line 266-268) Thus, further researches are warranted to find out effective methods to reduce body iodine pool in case of rhTSH use. Additionally, the efficacy of such methods on outcomes of RAI treatment needs to be evaluated.

(Page 14, line 296-299) Finally, differences in outcomes of ablation or treatment were not analyzed in this study. Because the main purpose of this study was to compare LID efficacy between two different TSH stimulation protocols, all repeated scans and RAI treatments were included in this study.

8. The abstract would need re-writing to reflect these comments.

We rewrote the abstract.

Minor

[Introduction]

1. Define intermediate and high-risk thyroid cancer and high-dose 131I.

Because the definition criteria of each risk group is a somewhat long list, please refer to the reference paper of ATA guidelines (Ref. 1). We added a usual definition for high-dose 131I (� 3.7 GBq) as shown below. However, in our study we classified RAI treatments with a cutoff of 1.2 GBq.

(Page 2, line 57) high-dose 131I treatment (usually, � 3.7 GBq)

2. Line 47-49 (Additionally, the thyroid globulin level when thyroid stimulating hormone (TSH) is stimulated is the most sensitive biomarker for differentiated thyroid cancer.) Remove, this observation is not relevant to this paper. 

We revised the manuscript according to the reviewer’s comment.

3. It would be helpful to set out the research question(s) much more clearly and to provide some explanation as to why the analyses of other clinical factors were undertaken.

We revised the manuscript according to the reviewer’s comment.

(Page 2-3, line 74-76) The aim of the present study is to evaluate the influence of the different TSH stimulation protocols on the adequacy of LID. The influence of various clinical factors on the adequacy of LID were also investigated.

[Methods]

4. Please provide a full description of the inclusion and exclusion criteria.

We rephrased the manuscript to clearly describe the inclusion and exclusion criteria.

(Page 3, line 83-86) The inclusion criteria were as follows; patients with thyroid cancer who underwent LID for 131I treatment or 123I scan between 2016 and 2020, and available UICR measurements conducted for evaluating the adequacy of LID.

(Page 3, line 96-97) The exclusion criteria were as follows; UICR measurements of children less than 10-year-old at the time of measurement, and outlier results > 1000 �g/g Cr.

5. There are substantial variations in the LIDs advised between different countries and institutions. It would be useful to include a table showing the advised LID used in this study.

According to the reviewer’s comment, we added a new table to show LID instruction of our institution.

(Table 1) Instructions of LID for RAI scan or treatment

6. Statistical significance is a function of sample size – it is better to give the precise p-values (which has been done here), to talk about levels of evidence instead of referring to a result as ‘significant’ and to consider the effect size (see https://www.ncbi.nlm.nih.gov/pmc/articles/PMC1119478/)

In this manuscript, we decided to report P-values to 3 decimal places. Thus, the limit of reported P-value was 0.001. Almost all P-values were < 0.001 in the result, and one P-value that was larger than 0.001 was 0.013 and we reported the precise P-value for it.

[Results]

7. Lines 1136 - 116: Cut text from 985 to 2 times. All of this is presented in table 1. 

According to the reviewer’s comment, we revised the manuscript to remove redundant description. The numbers of patients who underwent LID once or � 2 times during the study period is different from the numbers shown in the Table.

(Page 6, line 140-142) 1045 (99.1%) were differentiated thyroid cancer, and 9 (0.9%) were poorly differentiated thyroid cancer. During the study period, 502 (47.6%) patients underwent LID once and 552 (52.4%) patients underwent LID more than once.

8. Line 129, 132, 133, 136. Quote the effect not the p-values in the text. For example, line 129 “There was strong evidence that UICR was higher in women than in men (55.0 (52.8) vs 40.4 (47.1) ug/gCr)” is much more meaningful than just quoting the p-value.

The values are shown in Table 2 in the previous manuscript. However, we revised the Table (now Table 4) and added the values in the text.

(Page 8, line 181-182; Page 8-9, line 184-186, 188-189) UICR was significantly higher in women than in men (55.0 � 52.8 vs. 40.4 � 47.1, P < 0.001, Table 4). … UICR was increased with age and there were significant differences between the age groups (35.0 � 31.1, 45.9 � 47.2, 56.1 � 50.3, and 83.3 � 86.6 for each age group, respectively, P < 0.001, Table 4). … UICR was significantly higher in the first LID than that in second or more LID (55.3 � 44.2 vs. 46.6 � 56.6, P < 0.001).

9. Retitle table 2 and the corresponding heading ‘according to demographic and clinical factors’.

We revised the title of the table (now, Table 4).

(Table 4) UICR according to demographic and clinical factors

10. Consider removing the figures and putting the information into table 2.

We understand the reviewer’s concern. However, we believe the figures are helpful for readers to easily see the results, and we want to show the figures.

11. The title of table 3 doesn’t match what the table is showing. Also, it needs to be clearer that the numbers in each cell in table 3 refers to numbers of patients.

We revised the title of Table 3, and also its format.

(Table 3) The number of patients according to TSH stimulation protocols and LID procedures in those who underwent the first and second LID during the study period

[Discussion]

12. Please start the discussion with the main findings, not information about whether urine iodine creatinine is a good measure (that can go under limitations).

We revised the manuscript according to the reviewer’s comment.

(Page 11, line 215-223) In the present study, UICR was analyzed in a large cohort including 1715 measurements of 1054 patients, and it was demonstrated that UICR is significantly associated with TSH stimulation protocol. In most cases (92.7%), 2-week LID resulted in ‘excellent’ or ‘adequate’ range of UICR. However, UICR was higher and proportion of ‘excellent’ adequacy was lower in patients who used rhTSH without THW compared to those used THW. In patients who underwent repeated LID using both rhTSH and THW protocols, UICR was higher in case of rhTSH, regardless of the order of used protocols. Furthermore, TSH stimulation protocol was a significant and independent determinant for achieving ‘excellent’ adequacy of LID.

13. “….body retention of RAI increases with use of LID”. This effect has been observed elsewhere (Maruca et al., 1984) and is not desirable (although Plujimen et al do not consider it to be highly problematic). Is this what the authors wanted to communicate from this paper?

We appreciate the reviewer’s comment. We intended to mean increased retention of RAI in remnant thyroid or cancer tissue as reported by Pluijmen et al. We revised it as follows, so that the meaning is clear.

(Page 12, line 230-231) such as low urine iodine level and high 24-hour neck RAI uptake in LID [12].

14. Please amend to show the study findings more completely ie that success rate was similar (about 80%) for all patients with UICR <250mg/gCr, not just those <50mcg/gCr. Sohn et al cannot be used to suggest that a UICR <50mcg/gCr is superior to <100mcg/gCr.

We revised the manuscript according to the reviewer’s comment.

(Page 12, line 231-234) In a relatively recent study, ablation success rate was 50% in patients with UICR � 250 �g/g Cr whereas it was 76–82% in patients with UICR < 50 �g/g Cr, although the ablation success rates of those with UICR < 250 �g/g Cr and those with < 50 �g/g Cr were similar [13].

15. Line 211 The European Thyroid Cancer Taskforce made that recommendation in 2006 and it does not reflect guidelines from other European bodies which mainly advise 1-2 weeks (eg European Association of Nuclear Medicine in 2008 recommended 1-2 weeks (Luster et al) – please add this reference. (And Pluijmen et al, cited as a study that demonstrated LIDs improve ablation only used 4 days – perhaps comment on that).

We revised the manuscript according to the reviewer’s comment. The 4-day protocol used by Pluijmen et al. has been cited in Line 231. 

(Page 12, line 242-244) While the American Thyroid Association and the European Association of Nuclear Medicine recommend 1 to 2-week LID prior to RAI administration,

16. Line 241 – 252. There is some discussion of the demographic and clinical differences, can the authors comment on whether changes to how dietary advice is delivered locally could help with these differences.

It may be considered to give intense education or instruction to old-aged patients or those who are scheduled for 131I treatment. However, because the main finding of this study is to compare those with rhTSH use and THW, we suppose it would be better not to include this comment in the manuscript.

17. I have not extensively proof-read the article or corrected minor errors in English but there are some and I suggest a proof reader be employed 

We appreciate the reviewer’s comment. We have requested additional proof-reading and corrected several points of errors.

---

## [Decision Letter · Decision Letter 1]

16 Aug 2021

Effect of TSH stimulation protocols on adequacy of low-iodine diet for radioiodine Administration

PONE-D-21-02040R1

Dear Dr. Paeng,

We’re pleased to inform you that your manuscript has been judged scientifically suitable for publication and will be formally accepted for publication once it meets all outstanding technical requirements.

Kind regards,

Ali S. Alzahrani

Academic Editor

PLOS ONE

Additional Editor Comments (optional):

Reviewers' comments:

Reviewer's Responses to Questions

**Comments to the Author**

1. If the authors have adequately addressed your comments raised in a previous round of review and you feel that this manuscript is now acceptable for publication, you may indicate that here to bypass the “Comments to the Author” section, enter your conflict of interest statement in the “Confidential to Editor” section, and submit your "Accept" recommendation.

Reviewer #1: All comments have been addressed

Reviewer #2: All comments have been addressed

2. Is the manuscript technically sound, and do the data support the conclusions?

Reviewer #1: Yes

Reviewer #2: Yes

3. Has the statistical analysis been performed appropriately and rigorously? 

Reviewer #1: Yes

Reviewer #2: Yes

4. Have the authors made all data underlying the findings in their manuscript fully available?

Reviewer #1: Yes

Reviewer #2: Yes

5. Is the manuscript presented in an intelligible fashion and written in standard English?

Reviewer #1: Yes

Reviewer #2: Yes

6. Review Comments to the Author

Reviewer #1: This is a well-deigned and well-written retrospective analysis in a large cohort to analyze effect of different clinical factors including the two TSH stimulation protocol on the LID utilizing the spot UICR.

Purpose of the study is important and impactful taking into consideration the increasing use of rhTSH and more practical spot UICR. Methodology is systematic, well-designed and clear; moreover, it can be replicated in future analysis to find out the impact on recurrence and survival that were not addressed in this study. Discussion is comprehensive and it covers the relevant literature, strength and weakness of current manuscript and the potential future directions and impact of current findings. Conclusion is well drawn based on the presented interesting data. However, it will be highly impactful if data on survival presented or to consider this by authors in future follow up work.

Reviewer #2: All my comments have been addressed, thank you.

7. PLOS authors have the option to publish the peer review history of their article (what does this mean?). If published, this will include your full peer review and any attached files.

Reviewer #1: **Yes: **Akram Al-Ibraheem

Reviewer #2: No

---

## [Editor Report · Acceptance letter]

27 Aug 2021

PONE-D-21-02040R1 

Effect of TSH stimulation protocols on adequacy of low-iodine diet for radioiodine Administration

Dear Dr. Paeng:

I'm pleased to inform you that your manuscript has been deemed suitable for publication in PLOS ONE. Congratulations! Your manuscript is now with our production department. 

Kind regards, 

on behalf of

Dr. Ali S. Alzahrani 

Academic Editor

PLOS ONE